# Transplantation of enteric nervous system stem cells rescues nitric oxide synthase deficient mouse colon

Conor J. McCann[1], Julie E. Cooper[1], Dipa Natarajan[1], Benjamin Jevans[1], Laura E. Burnett[1], Alan J. Burns[1,2] & Nikhil Thapar[1]

Enteric nervous system neuropathy causes a wide range of severe gut motility disorders. Cell replacement of lost neurons using enteric neural stem cells (ENSC) is a possible therapy for these life-limiting disorders. Here we show rescue of gut motility after ENSC transplantation in a mouse model of human enteric neuropathy, the neuronal nitric oxide synthase ($nNOS^{-/-}$) deficient mouse model, which displays slow transit in the colon. We further show that transplantation of ENSC into the colon rescues impaired colonic motility with formation of extensive networks of transplanted cells, including the development of $nNOS^+$ neurons and subsequent restoration of nitrergic responses. Moreover, post-transplantation non-cell-autonomous mechanisms restore the numbers of interstitial cells of Cajal that are reduced in the $nNOS^{-/-}$ colon. These results provide the first direct evidence that ENSC transplantation can modulate the enteric neuromuscular syncytium to restore function, at the organ level, in a dysmotile gastrointestinal disease model.

[1] Stem Cells and Regenerative Medicine, UCL Great Ormond Street Institute of Child Health, 30 Guilford Street, London WC1N, UK. [2] Department of Clinical Genetics, Erasmus Medical Center, Rotterdam 3015 CN, The Netherlands. Correspondence and requests for materials should be addressed to N.T. (email: n.thapar@ucl.ac.uk).

Neuropathological loss within the enteric nervous system (ENS) has been implicated in a wide range of severe gut motility disorders, such as achalasia[1–3], gastroparesis[4,5], slow transit constipation[6–8] and Hirschsprung's disease (aganglionic megacolon)[9,10], as well as being associated with a number of central nervous system disorders[11–13]. Potential replacement of lost neurons using stem cell replacement is an attractive therapy for such life-limiting disorders. Enteric neural stem cells (ENSC), which exist in both embryonic and adult gut, have been suggested as potential cell source for such treatments[14,15]. We and others have previously demonstrated the potential of both mouse[16,17] and human[18] ENSC to integrate within wild-type ganglionated mouse colon. Yet a limiting factor in the advancement of ENSC therapies for human application has been the failure to demonstrate functional rescue of motility in pathological disease models. Recent studies have demonstrated the successful integration of murine and human ENSC within aganglionic colon both *in vivo*[16,19] and *ex vivo*[20,21]; however, the severity of the gut phenotype and poor survival of homozygote mice has limited their *in vivo* use for investigating the potential functional rescue, at the organ level, of ENSC-based therapies.

Other models of neuronal loss are, therefore, essential to test the viability of cell-based transplantation techniques to restore functional deficits resulting from neuropathology. The loss of neuronal nitric oxide synthase (nNOS) has been implicated in a range of human enteric neuropathies[22], including oesophageal achalasia[23], infantile hypertrophic pyloric stenosis[24], gastroparesis (idiopathic and diabetic)[25], colonic dysfunction[26] and Hirschsprung's disease[27,28]. Notably, $nNOS^{-/-}$ mice recapitulate the clinical phenotype of a number of human diseases displaying both delayed gastric emptying[29–31], and slow transit in the colon[32] hence providing an ideal model to establish if ENSC can restore function after *in vivo* transplantation. Here we show rescue of motility, after transplantation of ENSC, within the $nNOS^{-/-}$ mouse colon. We further demonstrate robust restoration of nitrergic responses coincident with the development of nNOS$^+$ neurons in an nNOS-deficient microenvironment. In addition, we show concurrent rescue of interstitial cells of Cajal (ICC) within the $nNOS^{-/-}$ colon after ENSC transplantation. Thus, we propose that ENSC can modulate the neuromuscular syncytium via both cell-autonomous and non-cell-autonomous mechanisms to restore function, at the organ level, and ultimately rescue motility.

## Results

**Transplanted ENSC extensively integrate in $nNOS^{-/-}$ colon.**
To isolate ENSC, we used donor $Wnt1^{cre/+};R26R^{YFP/YFP}$ mice (P2–P7), in which neural crest cells and their enteric derivatives express endogenous yellow fluorescent protein (YFP). This endogenous expression allowed for isolation and fate-mapping of labelled donor ENSC. Selected YFP$^+$ cells maintained expression and formed characteristic neurospheres within 1 month in culture (Supplementary Fig. 1). To assess the composition of neurospheres, immunohistochemistry and qRT–PCR were performed to establish the presence of typical ENS cell types. Such neurospheres were found to express ENS markers such as the pan-neuronal marker TuJ1 (Supplementary Fig. 1a), the neural crest progenitor marker SOX10 (Supplementary Fig. 1b) and the glial marker S100 (Supplementary Fig. 1c). Notably, in addition to multipotent neural crest progenitors, neuronal markers, including NOS$^+$ neurons (Supplementary Fig. 1e–g), were observed within neurospheres *in vitro*. Hence, we sought to establish the ability of such neurospheres to colonize and populate $nNOS^{-/-}$ colon *in vivo*. As opposed to wild-type colon, which contains nNOS$^+$ cell bodies and fibres (Fig. 1a),

$nNOS^{-/-}$ mice display complete loss of nNOS$^+$ neurons in the colon (Fig. 1b).

We transplanted three YFP$^+$ neurospheres ($\sim 6 \times 10^4$ cells in total) into the distal colon of $nNOS^{-/-}$ mice at P14–P17 via laparotomy. Live imaging analysis, 4 weeks after transplantation, revealed the presence of extensive anastomosing networks of transplanted YFP$^+$ cells colonizing, on average, $5.46 \pm 0.5 \, mm^2$ ($n = 10$) of distal colon at the site of transplantation (Fig. 1c). Subsequent immunohistochemistry revealed more extensive networks of GFP$^+$ cells (Supplementary Figs 2 and 3). GFP$^+$ filamentous networks could be observed extending in both oral and aboral directions from the site of transplantation (Supplementary Fig. 2 and Supplementary Movie 1) including integration within the proximal colon. Post-acquisition mapping of transplanted cells revealed the largest continuous GFP$^+$ network extending 10.79 mm (Supplementary Fig. 2b). Along the length of the colon GFP$^+$ cells were found to co-express the neuronal marker TuJ1 (Fig. 1d–f) and project fibres (Fig. 1d, arrowheads), which contacted the endogenous neuronal network at the level of the myenteric plexus. GFP$^+$ cells were also identified within endogenous myenteric ganglia (Fig. 1g–i), where fine GFP$^+$ fibres and varicosities were observed encompassing and tracing the path of endogenous neuronal fibre tracts (Fig. 1g, arrowheads). Confocal imaging of the entire colon also revealed GFP$^+$ cells co-expressing TuJ1 integrated within ganglia along the length of the colon (Supplementary Fig. 3 and Supplementary Movie 2) up to a maximum of 42.4 mm from the site of transplantation thus confirming the ability of transplanted cells to migrate within the *tunica muscularis*. Transplanted cells displayed enteric neuronal characteristics including integration of bipolar (Supplementary Fig. 4a–d) and multipolar GFP$^+$ cells (Supplementary Fig. 4e,f) phenocopying the morphology of enteric interneurons and motor neurons, respectively.

**Transplanted ENSC regenerate nNOS$^+$ neurons.** To determine if transplanted ENSC have the capacity to develop an NOS$^+$ phenotype *in vivo* similar to *in vitro* cultures, immunohistochemistry and RT–PCR were performed. Within the distal colon, transplanted YFP$^+$ cells co-expressed both the neuronal marker TuJ1 and the neuronal nitric oxide synthase marker nNOS (Fig. 2a–d). nNOS$^+$ neurons were identified within ganglia-like structures (Fig. 2c,d, arrowheads) extending multiple nNOS$^+$ projections (Fig. 2c,d, arrows) within the network of transplanted cells. The presence of nNOS$^+$ neurons was further confirmed with PCR analysis demonstrating the specific expression of the nNOS transcript within transplanted colon compared with the complete absence of transcript in non-transplanted tissues (Fig. 2e).

To assess the proliferative capacity of transplanted cells, BrdU was applied 24 h post surgery and incorporation was assessed at 4 weeks. Incorporation of BrdU was observed within transplanted cells co-expressing TuJ1 (Fig. 2f–i, arrows) or nNOS (Fig. 2j–m, arrows) suggesting that transplanted ENSC have the ability to proliferate at early post-transplantation stages and subsequently differentiate to form mature neurons including nNOS$^+$ neurons within an nNOS-deficient microenvironment.

**Restoration of nitrergic responses in the $nNOS^{-/-}$ colon.** Having demonstrated the ability of ENSC to form nNOS$^+$ neurons *in vivo*, we assessed whether this integration could restore nitrergic responses in the $nNOS^{-/-}$ distal colon. On electrical field stimulation (EFS), C57BL/6J distal colon displayed large EFS-induced relaxations of $-1.98 \pm 0.15 \, g \, s$; $n = 5$ (Fig. 3a,e). By contrast loss of nNOS within $nNOS^{-/-}$ mice resulted in significant loss of this relaxatory response

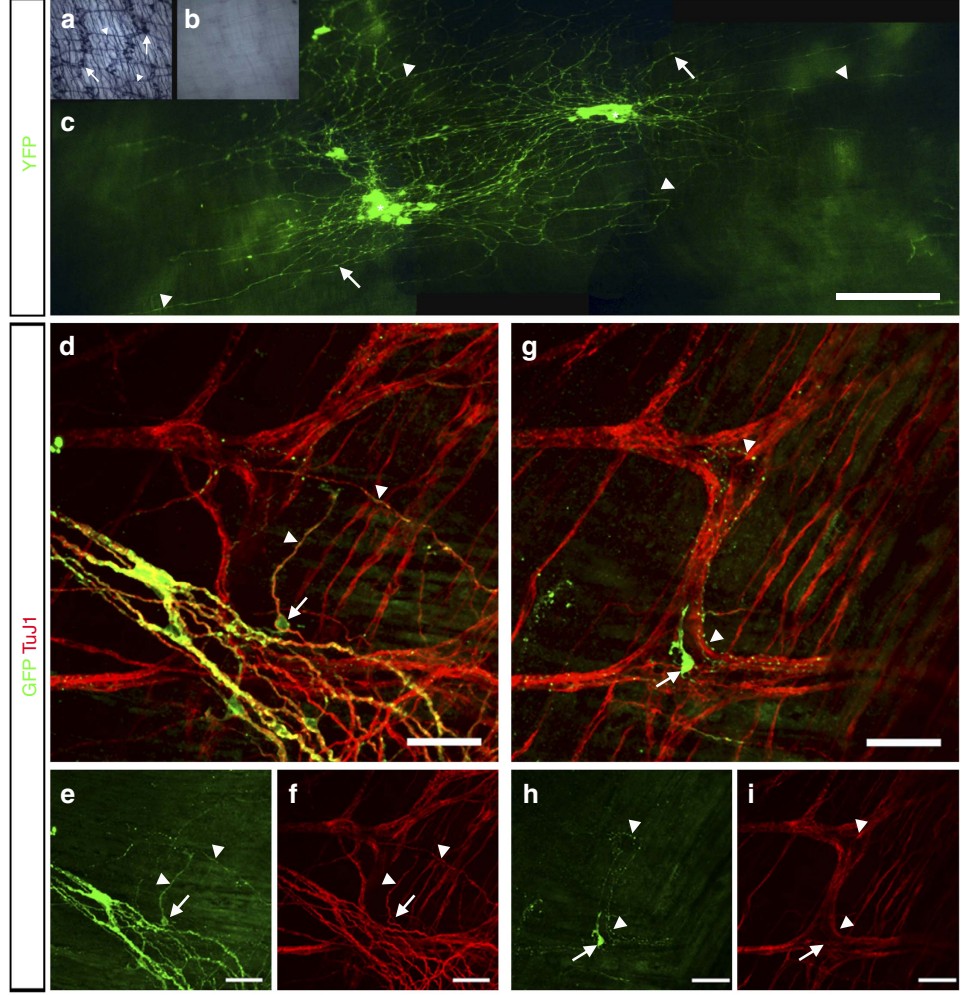

**Figure 1 | Transplanted ENSC extensively colonize and integrate within the *nNOS*$^{-/-}$ mouse colon. (a)** Representative image of NADPH diaphorase staining in wild-type C57BL/6J. nNOS$^+$ cell bodies (arrows), within enteric ganglia, and nNOS$^+$ fibres (arrowheads) are indicated. **(b)** Representative image of NADPH diaphorase staining of *nNOS*$^{-/-}$ mouse colon demonstrating loss of nitrergic innervation with the absence of NADPH diaphorase staining. **(c)** Representative stereoscopic montage image of transplanted YFP$^+$ cells within the *nNOS*$^{-/-}$ distal colon at 4 weeks. Transplanted cells form large anastomosing networks comprising cell bodies (arrows) and fibres (arrowheads), including ganglia-like structures (asterisk). Scale bar, 500 μm. **(d)** Example confocal image demonstrating co-expression of the neuronal marker TuJ1 within YFP$^+$-transplanted cells. YFP$^+$ cell bodies (arrow) project fibres (arrowheads) towards the endogenous neuronal network at the level of the myenteric plexus. **(e,f)** Individual channels showing GFP and TuJ1 staining from **d**. **(g)** Representative confocal image demonstrating the presence and integration of transplanted cells within endogenous ganglia. At the level of the myenteric plexus, fine GFP$^+$ fibres and varicosities encompass and trace endogenous neuronal fibre tracts (arrowheads). GFP$^+$ cells were also identified within ganglia (arrow). **(h,i)** Individual channels taken from **g** showing GFP and TuJ1 staining at the level of the myenteric plexus. Scale bar, 50 μm (**d-i**).

($-0.31 \pm 0.08$ g s; $n = 5$, $P < 0.0001$, Student's $t$-test; Fig. 3b,e). Sham-operated *nNOS*$^{-/-}$ distal colon displayed similarly abrogated EFS-induced responses ($-0.296 \pm 0.05$ g s; $n = 5$; Fig. 3c,e) to control *nNOS*$^{-/-}$ distal colon ($P = 0.8376$, Student's $t$-test). Interestingly, after ENSC transplantation there were statistically significant differences in mean values between control *nNOS*$^{-/-}$, sham-operated and transplanted *nNOS*$^{-/-}$ groups as determined by one-way analysis of variance (ANOVA) ($F(2,12) = 20.78$, $P \le 0.001$) with transplanted *nNOS*$^{-/-}$ distal colon displaying significant increases in EFS-induced relaxation ($-1.13 \pm 0.16$ g s; $n = 5$; Fig. 3d,e) compared to sham-operated *nNOS*$^{-/-}$ mice ($P = 0.0012$, Student's $t$-test). There was, however, a statistical difference between C57BL/6J distal colon and that of transplanted *nNOS*$^{-/-}$ mice ($P = 0.0034$, Student's $t$-test). On EFS, C57BL/6J distal colon also displayed post-stimulation rebound contractions with a mean amplitude of $0.326 \pm 0.04$ g; $n = 5$ (Fig. 3a,f). Similarly, *nNOS*$^{-/-}$ mice

displayed these post-stimulation rebound contractions ($0.22 \pm 0.06$ g; $n = 5$, $P = 0.1586$, Student's $t$-test; Fig. 3b,f). Sham-operated *nNOS*$^{-/-}$ distal colon displayed similar responses ($0.315 \pm 0.04$ g; $n = 5$; Fig. 3c,f) to *nNOS*$^{-/-}$ mice ($P = 0.2083$, Student's $t$-test). Interestingly, after ENSC transplantation, there were statistically significant differences in mean values between control *nNOS*$^{-/-}$, sham-operated and transplanted *nNOS*$^{-/-}$ groups by one-way ANOVA ($F(2,12) = 7.61$, $P = 0.007$) with transplanted *nNOS*$^{-/-}$ distal colon displaying significant increases in rebound contraction amplitude ($0.708 \pm 0.15$ g; $n = 5$; Fig. 3d,f) compared to sham-operated *nNOS*$^{-/-}$ mice ($P = 0.032$, Student's $t$-test) and C57BL/6J distal colon ($P = 0.0354$, Student's $t$-test).

To establish if the restored relaxatory response was due to the presence of transplanted nNOS$^+$ neurons, EFS-induced responses observed in non-adrenergic non-cholinergic conditions were analysed in the presence and absence of the nitric oxide

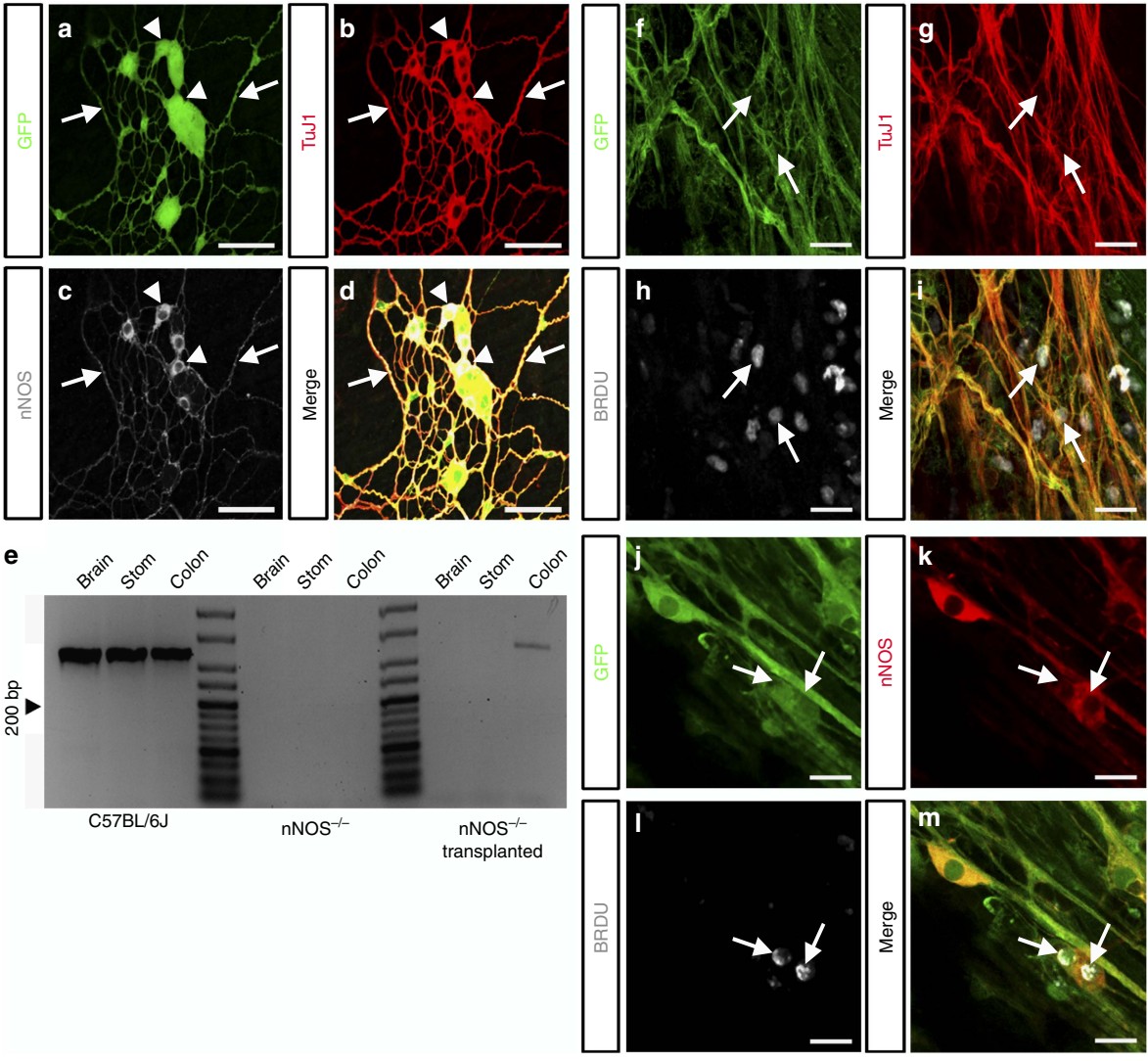

**Figure 2 | Transplanted ENSC form nNOS$^+$ neurons within the $nNOS^{-/-}$ distal colon.** (**a–c**) Representative confocal z-stack images demonstrating individual channels for GFP (**a**), TuJ1 (**b**) and nNOS (**c**) within a transplanted cell network present in the $nNOS^{-/-}$ distal colon. (**d**) Merged image of individual channels shown in (**a–c**) demonstrating triple labelling of transplanted YFP$^+$ cells (green) co-expressing the neuronal marker TuJ1 (red) and the neuronal nitric oxide synthase marker nNOS (grey). Individual transplanted nNOS$^+$ neuronal cell bodies (arrowheads) extend multiple nNOS$^+$ projections (arrows) to form an anastomosing network of transplanted cells. Scale bar, 50 μm. (**e**) Representative PCR gel demonstrating specific nNOS expression within transplanted $nNOS^{-/-}$ colon. (**f–m**) Example confocal images of BrdU incorporation within transplanted cells 4 weeks after application. Triple labelled transplanted cells (green, **f,j**) co-express either TuJ1 (red, **g**) or nNOS (red, **k**) having incorporated BrdU (grey, **h,l**) after administration 24 h post surgery. Scale bar, 20 μm.

synthase blocker L-NAME. L-NAME significantly reduced this EFS-induced relaxation ($-0.74 \pm 0.17$ g s versus $-0.12 \pm 0.16$ g s; $n = 4$; $P = 0.0389$, Student's $t$-test) in transplanted $nNOS^{-/-}$ colon (Fig. 3g–i). Taken together, these results indicate that transplantation of ENSC and development of nNOS$^+$ neurons in the $nNOS^{-/-}$ distal colon results in partial restoration of nitrergic responses.

**ENSC transplantation increases basal contractile properties.** Organ bath physiology also revealed large amplitude basal contractions in transplanted $nNOS^{-/-}$ distal colonic segments (Fig. 3d). To investigate these responses, basal contractile patterns were recorded in both the distal and proximal colonic segments in control conditions (Krebs solution), and after the addition of individual neurotransmitter antagonists or tetrodotoxin (TTX).

Wild-type C57BL/6J distal colon displayed significantly larger amplitude basal contractions ($0.10 \pm 0.01$ g; $n = 5$) compared to $nNOS^{-/-}$ mice ($0.05 \pm 0.01$ g; $n = 5$; $P = 0.0029$, Student's $t$-test; Fig. 4a,b,i). The distal colon of sham-operated $nNOS^{-/-}$ mice was found to display similar contractions ($0.09 \pm 0.02$ g; $n = 5$; Fig. 4c,i) to that of control $nNOS^{-/-}$ mice ($P = 0.0592$, Student's $t$-test). One-way ANOVA analysis suggested that there were statistically significant differences in basal contractile amplitude ($F(2,12) = 14.43$, $P = 0.001$) between control $nNOS^{-/-}$, sham-operated and transplanted $nNOS^{-/-}$ mice with significant increases in the average contractile amplitude in transplanted $nNOS^{-/-}$ distal colon ($0.30 \pm 0.06$ g, $n = 5$) compared with sham-operated $nNOS^{-/-}$ ($P = 0.0096$, Student's $t$-test; Fig. 4c,d,i). Contractile frequency, however, did not appear to be affected as determined by one-way ANOVA ($F(2,12) = 2.96$, $P = 0.090$, Fig. 4j). In addition, these increased basal contractile properties were not significantly affected by addition of individual

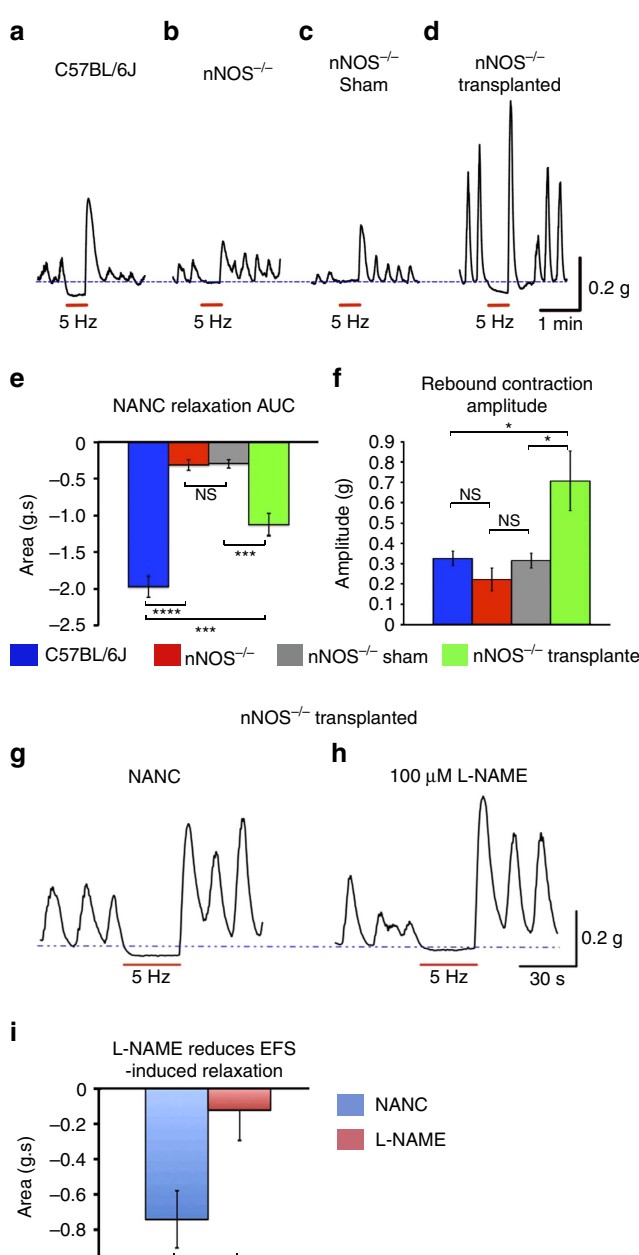

**Figure 3 | ENSC transplantation restores nitrergic responses in nNOS$^{-/-}$ distal colon.** (a–d) Representative traces of organ bath contractility in the presence of NANC demonstrating the response, in distal colonic segments, to EFS. (a) In response to EFS C57BL/6J distal colon demonstrates nitrergic relaxation, seen as relaxation below basal tension (dotted line). (b,c) nNOS$^{-/-}$ non-transplanted and sham-operated animals display a loss of this response. (d) Transplanted nNOS$^{-/-}$ mice display a partial restoration of the response at 4 weeks post transplantation. (e,f) Quantification of EFS-induced relaxation (area under curve, e) and rebound contraction (f) in C57BL/6J (blue bars, n = 5), nNOS$^{-/-}$ (red bars, n = 5), sham-operated nNOS$^{-/-}$ (grey bars, n = 5) and transplanted nNOS$^{-/-}$ (green bars, n = 5) distal colon. ****P ≤ 0.0001, **P ≤ 0.01, *P ≤ 0.05, by Student's t-test. (g–i) Representative traces of organ bath contractility in transplanted nNOS$^{-/-}$ colon in the absence or presence of the nitric oxide synthase antagonist L-NAME. (i) Summary data of the effect of L-NAME on the EFS-induced nitrergic response in transplanted nNOS$^{-/-}$ distal colon (n = 4). *P ≤ 0.05 by Student's t-test. Error bars represent mean ± s.e.m. in all panels.

neural antagonists or TTX (Fig. 4e–h and Supplementary Fig. 6) suggesting that transplantation of ENSC can lead to potential changes in underlying myogenic motility patterns.

To establish if ENSC transplantation can alter motility patterns beyond the transplanted distal colonic region, proximal colonic segments were investigated for changes in basal contractile patterns. Similar to the distal colon, wild-type C57BL/6J proximal colon displayed significantly larger amplitude contractions (0.34 ± 0.04 g; n = 3) compared to control nNOS$^{-/-}$ mice (0.10 ± 0.02 g; n = 3; P = 0.0044, Student's t-test; Supplementary Fig. 5a,b,i). Contractile amplitude in the proximal colon of sham-operated nNOS$^{-/-}$ mice was similar (0.10 ± 0.03 g, n = 3; P = 0.9931, Student's t-test) to that of nNOS$^{-/-}$ mice (Supplementary Fig. 5b,c,i). One-way ANOVA analysis suggested that there were statistically significant differences in basal contractile amplitude (F(2,6) = 7.38, P = 0.024) between control nNOS$^{-/-}$, sham-operated and transplanted nNOS$^{-/-}$ group means with transplanted nNOS$^{-/-}$ proximal colon displaying contractile events with significantly increased amplitude (0.32 ± 0.07 g; n = 3) compared with sham-operated nNOS$^{-/-}$ mice (P = 0.0412, Student's t-test; Supplementary Fig. 5c,d,i). Contractile frequency, however, did not appear to be affected as determined by one-way ANOVA (F(2,12) = 1.21, P = 0.362; Supplementary Fig. 5j). Thus, we conclude that ENSC transplantation to the distal colon results in changes in colonic motor patterns at sites distant from the transplanted region. Again, similar to the distal colon, these increased basal contractions were not affected by TTX (Supplementary Fig. 5e–h and Supplementary Fig. 6). The failure of TTX to attenuate these large amplitude colonic contractions in both the distal and proximal colon suggests that non-cell-autonomous changes in the neuromusculature occur post transplant.

**Transplantation decreases total intestinal transit time.** Previous studies have identified motility issues, including slow transit, in the colon of nNOS$^{-/-}$ mice. To assess the impact of ENSC transplantation, including the effect restored nitrergic responses and increased basal contractile properties would have on motility, total gastrointestinal (GI) transit time was measured. Control nNOS$^{-/-}$ mice displayed significantly prolonged transit time (177 ± 8.15 min; n = 5) compared to C57BL/6J mice (117.4 ± 2.64 min; n = 5; P = 0.0001, Student's t-test; Fig. 5a). Sham-operated nNOS$^{-/-}$ mice displayed similar transit time (185.2 ± 14.08 min; n = 5) to non-transplanted nNOS$^{-/-}$ controls (P = 0.63, Student's t-test). Notably, there were significant differences between control nNOS$^{-/-}$, sham-operated and transplanted nNOS$^{-/-}$ group means as determined by one-way ANOVA (F(2,12) = 16.02, P < 0.0001). Moreover, total GI transit time in transplanted nNOS$^{-/-}$ mice was significantly decreased (114.8 ± 3.60 min; n = 5; P = 0.0001, Student's t-test) compared with sham-operated nNOS$^{-/-}$ mice, reducing transit time towards levels observed in C57BL/6J mice (P = 0.5761, Student's t-test) suggesting rescue of motility (Fig. 5a).

Analysis of faecal output revealed a significant reduction in output in nNOS$^{-/-}$ mice (21.61 ± 3.71 mg h$^{-1}$; n = 5) compared with C57BL/6J mice (35.97 ± 2.63 mg h$^{-1}$; n = 5; P = 0.0134, Student's t-test; Fig. 5b). In addition, sham-operated nNOS$^{-/-}$ displayed similar faecal output (24.41 ± 2.41 mg h$^{-1}$; n = 5) to that of nNOS$^{-/-}$ controls (P = 0.544, Student's t-test). Interestingly, there were statistically significant differences in faecal output between control nNOS$^{-/-}$, sham-operated and transplanted nNOS$^{-/-}$ group means as determined by one-way ANOVA (F(2,12) = 11.51, P = 0.002) with transplantation of ENSC resulting in significant increases in faecal output in

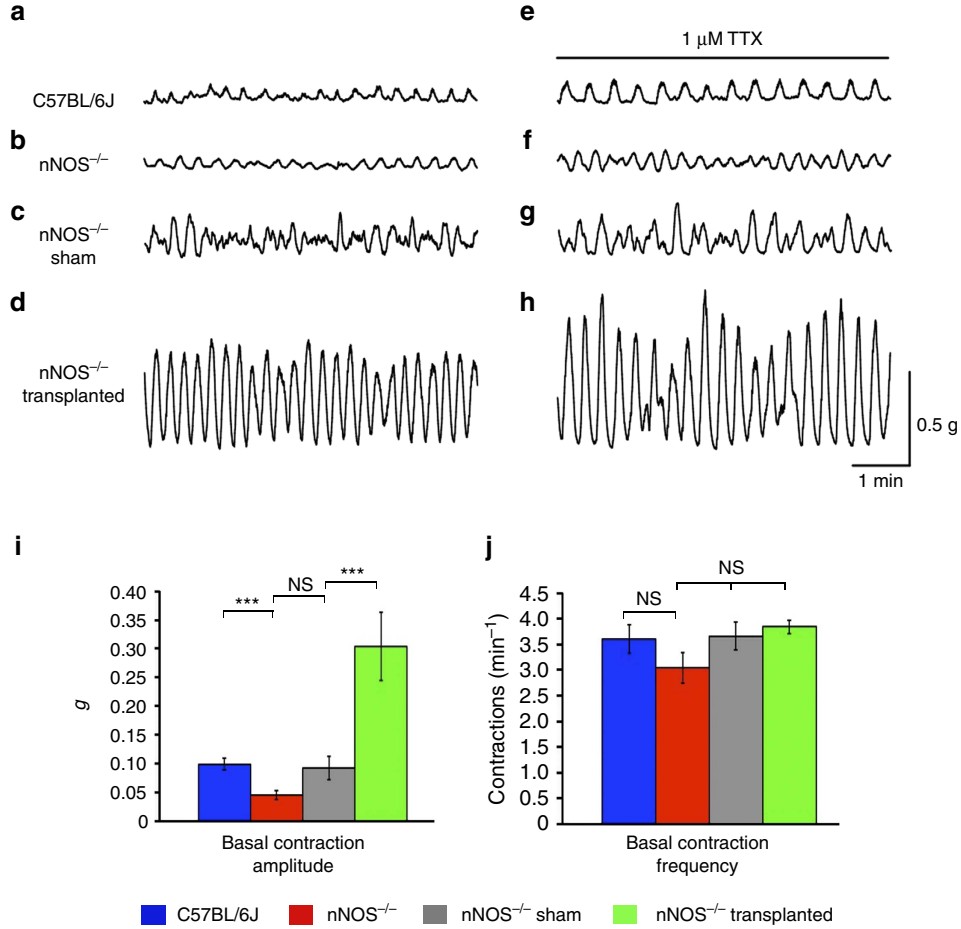

**Figure 4 | ENSC transplantation alters basal contractile patterns in *nNOS*$^{-/-}$ distal colon.** (**a**–**d**) Representative organ bath contractility traces demonstrating basal contractile properties of C57BL/6J, *nNOS*$^{-/-}$, sham-operated *nNOS*$^{-/-}$ and transplanted *nNOS*$^{-/-}$ distal colon in control conditions. (**e**–**h**) Representative organ bath contractility traces demonstrating basal contractile properties of C57BL/6J, *nNOS*$^{-/-}$, sham-operated *nNOS*$^{-/-}$ and transplanted *nNOS*$^{-/-}$ distal colon from **a**–**d** after the addition of TTX. (**i**) Quantification of basal contractile amplitude in C57BL/6J (blue bars), *nNOS*$^{-/-}$ (red bars), sham-operated *nNOS*$^{-/-}$ (grey bars) and transplanted *nNOS*$^{-/-}$ (green bars) distal colon. $n = 5$ for each group. ***$P \leq 0.001$ by Student's $t$-test. (**j**) Quantification of basal contractile frequency in C57BL/6J (blue bars), *nNOS*$^{-/-}$ (red bars), sham-operated *nNOS*$^{-/-}$ (grey bars) and transplanted *nNOS*$^{-/-}$ (green bars) distal colon. $n = 5$ for each group. Error bars represent mean ± s.e.m. in all panels.

transplanted *nNOS*$^{-/-}$ mice ($44.95 \pm 4.77\,\mathrm{mg\,h^{-1}}$; $n = 5$) compared with sham-operated *nNOS*$^{-/-}$ mice ($P = 0.0049$, Student's $t$-test), restoring faecal output to levels comparable to C57BL/6J mice ($P = 0.1378$, Student's $t$-test).

**Upper GI transit is not affected by ENSC transplantation.** As *nNOS*$^{-/-}$ mice display pan-enteric deficits in nNOS signalling and have been shown to have delayed gastric emptying, we sought to assess if transplantation to the distal colon could affect intestinal transit parameters outside of the colonic region. Using fluorescent *in vivo* imaging, liquid stomach emptying at 30 min (Fig. 5c,d) was significantly delayed in *nNOS*$^{-/-}$ mice ($32.9 \pm 5.9\%$, $n = 3$) compared to C57BL/6J mice ($56.4 \pm 4.6\%$; $n = 3$; $P = 0.0316$, Student's $t$-test) similar to previously described studies. Notably, no difference was observed in liquid stomach emptying between control *nNOS*$^{-/-}$, sham-operated and transplanted *nNOS*$^{-/-}$ group means as determined by one-way ANOVA ($F(2,6) = 1.00$, $P = 0.421$; Fig. 5c,d).

To assess partial intestinal transit, mice were killed 90 min after gavage of a fluorescent dye. Fluorescent imaging was performed *ex vivo* and the distance the dye had transited was calculated as a

percentage of total intestinal length (Fig. 5e,f). *nNOS*$^{-/-}$ mice displayed reduced intestinal transit ($79.9 \pm 2.1\%$, $n = 3$) compared to C57BL/6J mice ($89.1 \pm 1.1\%$, $n = 3$; $P = 0.0176$, Student's $t$-test). Again, there was no difference between control *nNOS*$^{-/-}$, sham-operated and transplanted *nNOS*$^{-/-}$ transit distance at 90 min as determined by one-way ANOVA ($F(2,6) = 0.56$, $P = 0.6$). We conclude that as both stomach emptying and partial intestinal transit time are unaffected by transplantation, the overall improvement in total intestinal transit time in transplanted *nNOS*$^{-/-}$ mice is due to substantial increases in colonic transit.

**ENSC transplantation restores ICC numbers.** We next sought to ensure that the changes in GI function following transplantation were not secondary to other phenomena such as inflammation. On initial dissection, and after careful analysis, no significant changes in the gross anatomy of the colon or evidence of inflammatory responses were observed (Supplementary Fig. 7a). In addition, no inflammation was observed on histological examination (Supplementary Fig. 7b). Histological analysis also revealed that there were no differences in either colonic diameter ($P > 0.05$; Supplementary Fig. 7c,e) or muscle thickness

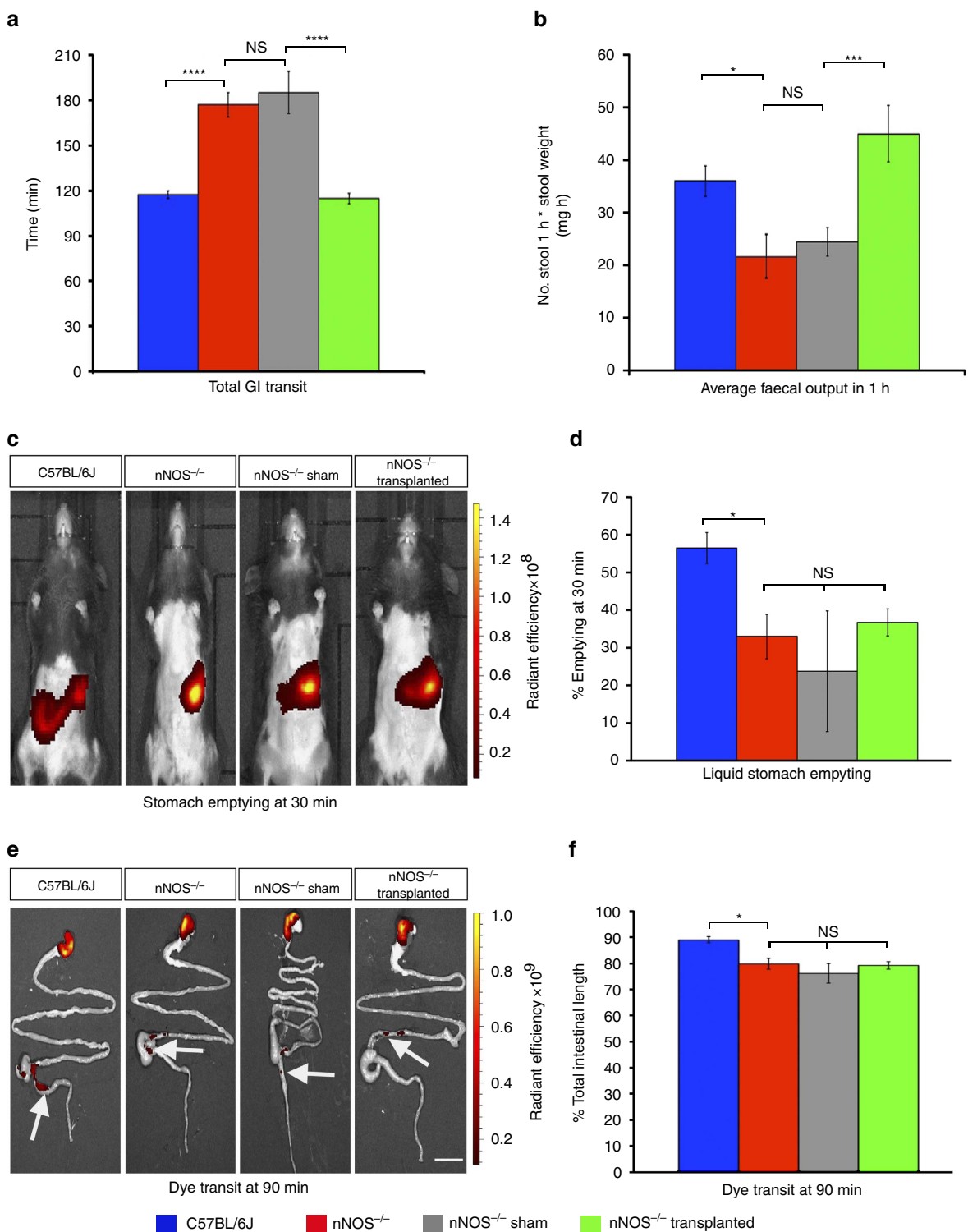

**Figure 5 | Analysis of GI transit time and stool production in transplanted *nNOS*$^{-/-}$ mice.** (**a**,**b**) Summary data of total GI transit time (**a**) measured after oral gavage of dye, and faecal output (**b**), 4 weeks after transplantation in C57BL/6J (blue bars), non-transplanted *nNOS*$^{-/-}$ (red bars), sham-operated *nNOS*$^{-/-}$ (grey bars) and transplanted *nNOS*$^{-/-}$ (green bars) mice. $n = 5$ for each group, ****$P \leq 0.0001$, ***$P \leq 0.001$, **$P \leq 0.01$, *$P \leq 0.05$ by Student's $t$-test. (**c**) Representative *in vivo* fluorescent images 30 min after gavage of a fluorescent marker. (**d**) Quantification of percentage stomach emptying at 30 min in C57BL/6J (blue bars), *nNOS*$^{-/-}$ (red bars), sham-operated *nNOS*$^{-/-}$ (grey bars) and transplanted *nNOS*$^{-/-}$ (green bars) mice. $n = 3$ for each group. *$P \leq 0.05$ by Student's $t$-test. (**e**) Example fluorescent images of the C57BL/6J, *nNOS*$^{-/-}$, sham-operated *nNOS*$^{-/-}$ and transplanted *nNOS*$^{-/-}$ intestine *ex vivo*, 90 min after gavage of a fluorescent marker. (**f**) Summary data of dye transit at 90 min in C57BL/6J (blue bars), *nNOS*$^{-/-}$ (red bars), sham-operated *nNOS*$^{-/-}$ (grey bars) and transplanted *nNOS*$^{-/-}$ (green bars) mice. $n = 3$ for each group. *$P \leq 0.05$ by Student's $t$-test. Error bars represent mean ± s.e.m. in all panels.

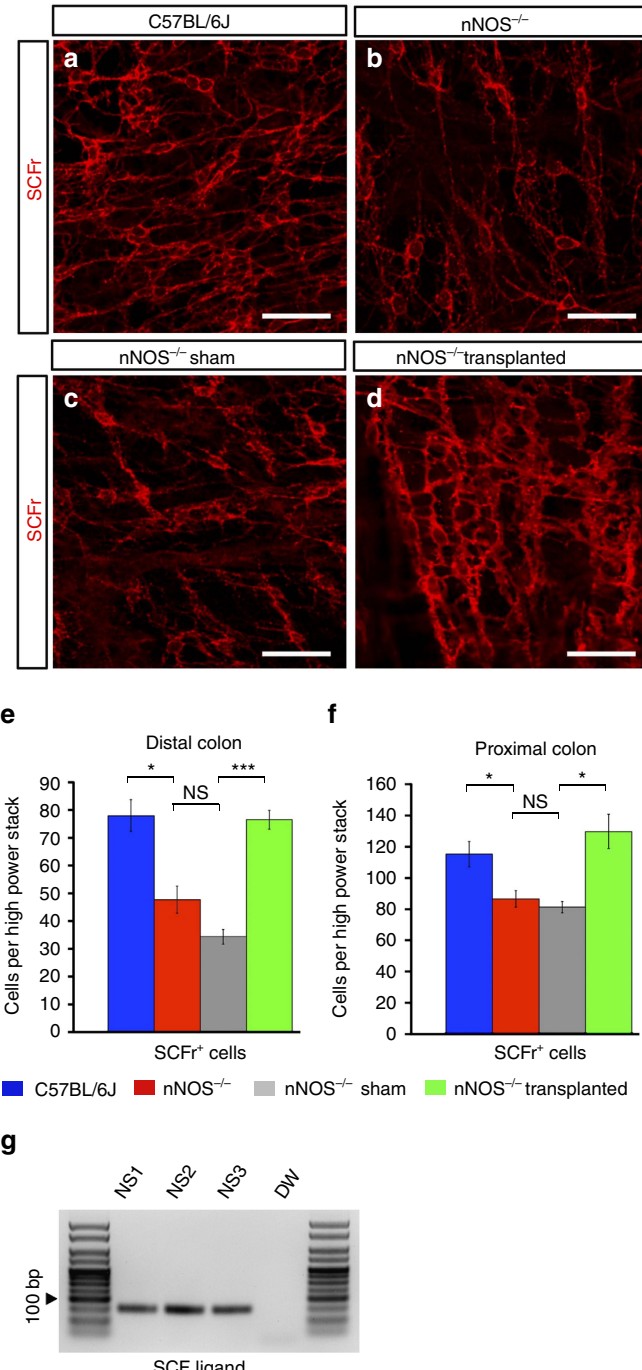

**Figure 6 | ENSC transplantation alters ICC networks in $nNOS^{-/-}$ colon.** (**a–d**) Representative z-stack confocal images of ICC in C57BL/6J, $nNOS^{-/-}$, sham-operated $nNOS^{-/-}$ and transplanted $nNOS^{-/-}$ distal colon. Scale bar, 50 μm. (**e,f**) Quantification of ICC numbers in the distal (**e**) and proximal (**f**) colon in C57BL/6J (blue bars), $nNOS^{-/-}$ (red bars), sham-operated $nNOS^{-/-}$ (grey bars) and transplanted $nNOS^{-/-}$ (green bars) mice calculated as a mean of five high-power stacks per region per mouse. $n = 3$ for each group. $***P \leq 0.001$, $P \leq 0.05$ by Student's $t$-test. Error bars represent mean ± s.e.m. in all panels. (**g**) Representative PCR gel demonstrating expression of SCF ligand in $Wnt1^{cre/+}$;$R26R^{YFP/YFP}$-derived neurospheres (NS). Neurosphere cultures were derived from separate $Wnt1^{cre/+}$;$R26R^{YFP/YFP}$ intestines ($n = 3$). Autoclaved Milli-Q water (DW) was used as a non-template negative control.

($P > 0.05$; Supplementary Fig. 7d,f) when comparing both the proximal and distal colon in $nNOS^{-/-}$, sham-operated $nNOS^{-/-}$ and transplanted $nNOS^{-/-}$ animals by one-way ANOVA.

To assess if ENSC transplantation may alter other cells involved in neuromuscular signalling and excitation–contraction coupling, the numbers of ICC were quantified in both the distal (Fig. 6a–e) and proximal (Fig. 6f and Supplementary Fig. 8) colon. Notably, ICC were reduced in both distal ($48 \pm 5$; $n = 3$; Fig. 6e) and proximal ($87 \pm 5$; $n = 3$; Fig. 6f) $nNOS^{-/-}$ colonic segments compared to C57BL/6J distal ($78 \pm 6$; $n = 3$; $P = 0.0158$, Student's $t$-test) or proximal colon ($115 \pm 8$; $n = 3$; $P = 0.0410$, Student's $t$-test). No difference was found between ICC numbers in either the distal ($34 \pm 3$; $n = 3$; $P = 0.0743$, Student's $t$-test) and proximal colon ($81 \pm 4$; $n = 3$; $P = 0.4504$, Student's $t$-test) in sham-operated $nNOS^{-/-}$ compared to control $nNOS^{-/-}$ mice (Fig. 6e,f). By contrast, one-way ANOVA analysis determined that there were statistically significant differences in ICC numbers in both the distal ($F(2,6) = 6.47$, $P = 0.032$) and proximal ($F(2,6) = 26.29$, $P = 0.001$) colon when comparing non-transplanted $nNOS^{-/-}$, sham-operated $nNOS^{-/-}$ and transplanted $nNOS^{-/-}$ mice. Of interest, increased ICC numbers were observed in transplanted $nNOS^{-/-}$ mice both in the distal ($77 \pm 3$; $n = 3$) and proximal colon ($130 \pm 11$; $n = 3$) compared to sham-operated $nNOS^{-/-}$ distal ($P = 0.0006$, Student's $t$-test; Fig. 6e) and proximal ($P = 0.0135$, Student's $t$-test; Fig. 6f) colon, respectively. This increase in ICC numbers associated with ENSC transplantation suggests indirect modulation of cells that contribute to the neuromuscular syncytium, alongside the integration of transplanted neural crest-derived cells. To further assess this indirect modulation, we investigated if ENSC express trophic factors that may influence ICC development. PCR analysis demonstrated the expression of the stem cell factor (SCF) ligand within ENSC neurospheres in vitro (Fig. 6g) providing a possible mechanism by which transplanted cells could alter ICC development and maintenance after transplantation.

## Discussion

Recent studies have demonstrated the functional integration of both mouse and human ENSC-derived neurons in wild-type mouse colon after in vivo transplantation[16–18]. Functional analysis in these animals that lack a neuropathological phenotype is restricted to individual or groups of transplanted neurons rather than assessment of colonic physiology. We now report that transplantation of a population of selected enteric neural crest-derived cells has a clear and positive functional impact, rescuing motility in a pathophysiological mouse model that recapitulates the phenotype of several clinically relevant human disorders. We demonstrate that this rescue is achieved through both cell-autonomous restoration of nitrergic responses that were absent, and non-cell-autonomous rescue of ICC numbers, which were also found to be deficient in the $nNOS^{-/-}$ colon.

The ability to form nitrergic neurons is a critical step in the development of 'normal' enteric circuitry and many enteric disorders would likely benefit from the transplantation and engraftment of $nNOS^+$ cells. In clearly demonstrating the potential to restore nNOS neurons after ENSC transplantation, which to our knowledge is the first study to show functional effects at the organ level, we believe ENSC therapies could impact widely at the clinical level. Interestingly, transplantation of ENSC led to restoration of nitrergic responses and increases in post-stimulation 'rebound' contraction. Previous studies have suggested that this post-stimulation contraction is mediated via

nitric oxide[33] or through a generalized inhibitory response via eicosanoids rather than linked to a distinct inhibitory transmitter[34]. The finding that transplantation of ENSC cells restored nitrergic inhibitory responses and altered post-stimulus responses suggests adaptation of neuromuscular signalling mechanisms at multiple levels. Of note, there are differences in the methodology of our study and that of these earlier studies, hence further investigation is required to assess the functional interaction of transplanted ENSC at the cellular level. Furthermore, the novel finding that ENSC transplantation rescues ICC numbers in the $nNOS^{-/-}$ colon in a non-cell-autonomous fashion raises interesting questions regarding the interaction of transplanted ENSC with associated cells within intestinal tissues. Intestinal excitation–contraction coupling is extremely complex, with GI smooth muscle receiving inputs from multiple excitable cells. ICC exist within extensive networks throughout the intestinal tract[35] exhibiting pacemaker activity[36–38] and transducing neural signals to the smooth muscle[39,40]. Recent studies show that ICC are innervated by nitrergic nerves[41] and express nitric oxide sensitive guanylate cyclase in both guinea pig[42] and mouse colon[43] and that disruptions in nNOS signalling can result in loss of ICC in the stomach[25,44]. The findings of similar disruptions in ICC networks, in the present study, along the length of the $nNOS^{-/-}$ colon further emphasize the link between NOS signalling and ICC development. Our previous work has demonstrated that the selection of ENSC, as in this study, excludes incorporation of mesenchyme-derived ICC[45]. Notably, previous studies have suggested that enteric neurons are the dominant source of SCF, the natural ligand for c-Kit[46], and that 60% of nNOS$^+$ enteric neurons express SCF[47] possibly providing a direct signalling link between enteric neurons and ICC. The finding that ENSC-derived neurospheres express SCF ligand in line with these earlier reports provides a possible mechanism whereby ICC modification post transplantation is likely through SCF signalling from transplanted neurons.

One of the challenges for cell replenishment therapies is scaling up for potential human application. It remains possible that significant cell numbers will be required to facilitate functional outcomes in human patients. A recent study has demonstrated derivation of enteric neural crest from human pluripotent stem cells and potential rescue of a Hirschsprung phenotype[48]. As opposed to our study, this investigation transplanted up to 4 million cells to $Ednrb^{s-l/s-l}$ (SSL/LEJ) colon. While transplantation led to survival of mice, no mechanism was presented of the graft-mediated host rescue. The findings of our study suggest that in addition to engraftment of neural crest cells, non-cell-autonomous modification of the neuromuscular apparatus may be responsible for this rescue. Our study thus highlights the potential to apply a limited number of cells, to a particular area, which subsequently could impact on function throughout the organ and have significant clinical benefits. In addition, we demonstrate the potential for collecting postnatal tissue for use in transplantation studies and the ability of cells from this source to restore function. This caveat has significant therapeutic benefits as ENSC could be collected autologously or from matched donors to limit the potential for immunological rejection. Moreover, previous long-term safety studies using identical postnatal ENSC have demonstrated long-term survival of ENSC-derived cells restricted only to the region of transplantation[16], which may provide a substantial benefit over the potential therapeutic application of cells derived from pluripotent sources.

We conclude that this study provides the first evidence that ENSC can rescue GI motility within a neuropathic model and may provide the basis for development of targeted cellular therapies for enteric neuropathies.

## Methods

**Animals.** Male and female $Wnt1^{cre/+};R26R^{YFP/YFP}$ mice, in which neural crest cells express YFP, were used as donors to obtain YFP$^+$ ENSC. Heterozygote nNOS (B6.129S4-$Nos1^{tm1Plh}$/J) mice were obtained from The Jackson Laboratory (Bar Harbor, MN, USA). Male and female homozygote nNOS knockout ($nNOS^{-/-}$) mice were bred and maintained for use as recipients. Four–5-week-old C57BL/6J mice were obtained from The Jackson Laboratory (Bar Harbor, MN, USA) and killed as age-matched controls at 6 weeks. Animals used for these studies were maintained, and the experiments performed, in accordance with the UK Animals (Scientific Procedures) Act 1986 and approved by the University College London Biological Services Ethical Review Process. Animal husbandry at UCL Biological Services was in accordance with the UK Home Office Certificate of Designation.

**Cell isolation and enrichment.** The entire small intestine and colon was obtained from early postnatal (P2–P7) $Wnt1^{cre/+};R26R^{YFP/YFP}$ mice after cervical dislocation, and removed to sterile PBS for further dissection. Jejunum, ileum and colon muscle strips were obtained following removal of the mucosa via fine dissection. Intestinal cells were dissociated and YFP$^+$ cells isolated using fluorescence-activated cell sorting with a MoFloXDP cell sorter (Beckman Coulter, UK). YFP positive (YFP$^+$) cells were selected using a 530/40 filter set. Gating parameters were set using cells from wild-type gut and applied to increase specificity of selection of YFP$^+$ cells.

**Neurosphere culture.** YFP$^+$ cells were plated on fibronectin-coated six-well dishes in 'neurosphere medium' (NSM; DMEM F12 supplemented with B27 (Invitrogen, UK), N2 (Invitrogen, UK), 20 ng ml$^{-1}$ EGF (Peprotech, UK), 20 ng ml$^{-1}$ FGF (Peprotech, UK) and Primocin antibiotic (InvivoGen, UK) and maintained in culture for up to 4 weeks. Typically such cultures from early postnatal (P2–P7) intestine formed 'neurospheres' at ∼1 week in culture.

***In vivo* ENSC transplantation.** YFP-expressing neurospheres derived from $Wnt1^{cre/+};R26R^{YFP/YFP}$ mice were transplanted into the distal colon of P14–P17 $nNOS^{-/-}$, via laparotomy under isoflurane anaesthetic. Briefly, the distal colon was exposed and a small pocket was created in the *tunica muscularis* with the bevel of a 30G needle. A neurosphere, containing ∼2 × 10$^4$ YFP$^+$ cells, was subsequently transplanted to this site by mouth pipette using a pulled glass micropipette. Each transplanted tissue typically received three neurospheres (∼6 × 10$^4$ YFP$^+$ cells in total). Transplanted $nNOS^{-/-}$ mice were typically maintained for 4 weeks post transplantation before killing and removal of the colon for analysis. 'Sham' operations were performed as controls in which the intestine was manipulated in an identical fashion without the addition of YFP$^+$ cells.

**RT–PCR.** Total RNA was isolated from C57BL/6J, $nNOS^{-/-}$ or transplanted $nNOS^{-/-}$ brain, stomach and colon using TRIzol reagent (Life Technologies Ltd, Paisley, UK) and treated with DNase I (Qiagen, Manchester, UK). First-strand cDNA was amplified from 1 μg RNA using SuperScript VILO cDNA Synthesis Kit (Life Technologies Ltd, Paisley, UK). PCR was performed using region-specific primers for *nNOS* (Supplementary Table 1) using HotStarTaq DNA Polymerase (Qiagen, Manchester, UK). PCR reactions were performed in a PTC-200 Peltier Thermal Cycler (MJ Research Inc. Waltham, MA, USA). The amplification profile was 95 °C for 3 min, 35 cycles of 94 °C for 30 s, 60 °C for 60 s and 72 °C for 30 s, followed by a final step of 72 °C for 2 min. RT–PCR amplification fragments were analysed on a 2% agarose gel alongside a Hyperladder 25 bp marker (Bioline, London, UK).

**qRT–PCR.** Total RNA was isolated from three pooled neurospheres at the time of surgery using an RNeasy Micro Kit (Qiagen, Hilden, Germany), according to the manufacturer's instructions. First-strand cDNA was amplified from 100 ng RNA using SuperScript VILO cDNA Synthesis Kit (Life Technologies Ltd, Paisley, UK). RT quantitative PCR was performed with an ABI Prism 7500 sequence detection system (Applied Biosystems) using the Quantitect SYBR Green PCR kit (Qiagen, Hilden, Germany) according to the manufacturer's instructions. qRT–PCR was performed in triplicate using region-specific primers for GAPDH, TuJ1, SOX10 and S100 (Supplementary Table 1). Gene expression data were expressed as a proportion of GAPDH housekeeping gene, as a reference, using a 1/ΔCt calculation.

**Immunohistochemistry.** Neurosphere immunohistochemistry was performed following paraformaldehyde fixation (4% w/v in 0.1 mol l$^{-1}$ PBS for 45 min at 22 °C). After fixation, neurospheres were washed for 1 h in PBS (0.01 mol l$^{-1}$, pH 7.2 at 4 °C). Neurospheres were blocked for 1 h (0.1 mol l$^{-1}$ PBS containing 1% Triton X-100, 1% BSA and 0.15% glycine) at 22 °C. The primary antibodies used in the study are listed in Supplementary Table 2. Intact neurospheres were incubated in primary antibody (diluted in 0.1 mol l$^{-1}$ PBS containing 1% Triton X-100, 1% BSA and 0.15% glycine) overnight at 4 °C and immunoreactivity was detected using secondary antibodies (1:500 in 0.1 mol l$^{-1}$ PBS, 1 h at room

temperature, Supplementary Table 3). Before mounting, neurospheres were washed thoroughly in PBS (0.1 mol l$^{-1}$ PBS for 1 h at 22 °C).

Whole-mount immunohistochemistry was performed on transplanted colonic segments after excision and removal of the mucosa by sharp dissection. Tissues were fixed in paraformaldehyde (4% w/v in 0.1 mol l$^{-1}$ PBS for 45 min at 22 °C). After fixation, tissues were washed for 24 h in PBS (0.01 mol l$^{-1}$, pH 7.2 at 4 °C). Tissues were blocked for 1 h (0.1 mol l$^{-1}$ PBS containing 1% Triton X-100, 10% sheep serum). Tissues were incubated in primary antibody (diluted in 0.1 mol l$^{-1}$ PBS containing 1% Triton X-100, 10% sheep serum, Supplementary Table 2) for 48 h at 4 °C and immunoreactivity was detected using the secondary antibodies listed in Supplementary Table 3 (1:500 in 0.1 mol l$^{-1}$ PBS, 1 h at room temperature). Before mounting, tissues were washed thoroughly in PBS (0.1 mol l$^{-1}$ PBS for 2 h at 22 °C). Control tissues were prepared by omitting primary or secondary antibodies. Tissues and neurospheres were examined using a LSM710 meta confocal microscope (Zeiss, Germany). Confocal micrographs were digital composites of the Z-series of scans (0.5 μm optical sections). For cell-counting experiments, five Z-series scans (21 × 1 μm optical sections including longitudinal muscle, myenteric plexus and circular muscle layers) were obtained in each region per animal. Z-series files were blinded before counting. Individual cells were identified with DAPI co-labelling and cell counts performed across each individual z-scan using the cell counter plugin (FIJI). Each Z-series count (21 sections) was summed and average cell number was calculated as a mean of five Z-series scans per region in each animal. For confocal montage experiments, tissues were examined using a LSM880 multiphoton microscope (Zeiss, Germany). Confocal micrographs of whole mounts were digital composites of the Z-series of scans stitched using Zen software (Zeiss, Germany). Final images were constructed using FIJI software[49] after applying a post-acquisition Gaussian filter with a σ of 2.

For NADPH diaphorase staining, colonic tissues were prepared, fixed and washed as above. NADPH diaphorase activity was detected by incubating tissues in 0.1 mol l$^{-1}$ PBS containing 0.05% Triton X-100, 1 mg ml$^{-1}$ β-NADPH (Sigma, UK) and 0.5 mg ml$^{-1}$ nitrobluetetrazolium (Sigma, UK) for 20 min at 37 °C. After staining, the tissues were washed thoroughly in PBS (0.1 mol l$^{-1}$) before mounting.

Haemeotoxylin and eosin colonic cryostat sections (20 μm) were obtained from frozen gelatin-embedded samples using a Leica CM1900 UV Cryostat (Leica Microsystems, UK) and processed for haemeotoxylin and eosin. Briefly, frozen colonic sections were post fixed in PFA (4% w/v in 0.1 mol l$^{-1}$ PBS for 45 min at 22 °C), washed thoroughly and haemeotoxylin solution (Harris modified; Sigma, UK) applied for 7 min at 22 °C. After washing, slides were immersed in acid alcohol (1% HCl in 70% EtOH) for 10 s and washed before application of 1% eosin Y (Fisher Scientific, UK) for 5 min at 22 °C. Samples were subsequently washed, dehydrated and cleared in Histoclear (2 × 1 min at 4 °C; National Diagnostics, UK) before mounting.

**In vivo transit analysis.** To test GI transit, either 100 μl Gastrosense 750 (Perkin Elmer, USA) or 100 μl Brilliant Blue FCT (E122) solution (Langdales, UK) was administered to the stomach via gavage at 6 weeks (4 weeks post transplantation). Total GI transit time was calculated from time of administration to the first visualization of dye in the stool. Stool output in 1 h was calculated as: total number of stool in 1 h × stool weight. To assess stomach emptying and partial transit time, Gastrosense 750 fluorescence was imaged using an IVIS Lumina III In Vivo Imaging System (Perkin Elmer, USA). At 30 min, in vivo images were obtained and percentage stomach emptying calculated as: $\left(1 - \left(\frac{\text{Stomach fluorescence}}{\text{Total flourescence}}\right)\right) \times 100$.

To assess partial intestinal transit, the GI tract (stomach to terminal colon) was removed 90 min after Gastrosense 750 administration. Flourescence images were obtained of the GI tract and percentage intestinal transit calculated as: $\frac{\text{Dye transit distance}}{\text{Total intestinal length}} \times 100$.

**Contractility.** Longitudinal colonic muscle strips were isolated and the mucosa removed by sharp dissection in oxygenated Krebs solution. Longitudinal muscle strips were mounted in tissue baths (10 ml, SI-MB4; World Precision Instruments Ltd, UK) connected via suture to force transducers (SI-KG20, World Precision Instruments Ltd, UK) under an initial tension of 0.5 g. Tissues were maintained at 37 °C with perfusion of oxygenated Krebs solution. Following a 60 min equilibration period, activity was recorded using a Lab-Trax-4 data acquisition system (World Precision Instruments Ltd, UK) in the absence and presence of non-adrenergic non-cholinergic conditions (Atropine; 1 μM, Phentolamine hydrochloride; 1 μM, Propranolol hydrochloride; 1 μM). Nerves were stimulated for 30 s (5 Hz; 40 V; 0.3 ms pulse duration) via EFS via platinum electrode loops placed at each end of the muscle strip using a MultiStim System (D330, World Precision Instruments Ltd, UK). Addition of L-NAME (100 μM) or TTX (1 μM) to the bath solution was used to assess the nitrergic and neurally mediated responses, respectively. EFS gave rise to neural responses that were sensitive to TTX. Data were collected, stored and analysed by computer using a data acquisition programme (Labscribe 2-1, World Precision Instruments Ltd, UK).

**Statistical analysis.** Data are expressed as mean ± s.e.m. Differences in the data were evaluated between $nNOS^{-/-}$ control, sham-operated and transplanted groups using one-way ANOVA and subsequent intergroup differences were determined by unpaired Student's $t$-test. $P$ values $< 0.05$ were taken as statistically significant. The '$n$ values' reported refer to the number of mice or colonic segments used for each protocol. Each muscle was taken from a separate animal.

**Data availability.** The authors declare that all data supporting the findings of this study are available within the article and its Supplementary Information files or from the corresponding author on reasonable request.

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

## Acknowledgements

The authors thank Dr Ayad Eddaoudi, Ms Stephanie Canning (UCL Great Ormond Street Institute of Child Health Flow Cytometry Facility) and Dr Dale Moulding (UCL Great Ormond Street Institute of Child Health Imaging Facility) for technical support. The authors also gratefully acknowledge use of the *In Vivo* Imaging services at the UCL Great Ormond Street Institute of Child Health, funded by The Alternative Hair Charitable Foundation. All research at Great Ormond Street Hospital NHS Foundation Trust and UCL Great Ormond Street Institute of Child Health are made possible by the NIHR Great Ormond Street Hospital Biomedical Research Centre. The views expressed are those of the author(s) and not necessarily those of the NHS, the NIHR or the Department of Health. This project has received some funding from the European Union's Horizon 2020 research and innovation programme 'INtestinal Tissue Engineering Solution' under grant No 668294. N.T. is supported by Great Ormond Street Hospital Children's Charity (GOSHCC—V1258). C.J.M., J.E.C. and D.N. were funded through a GOSHCC grant (W1018C) awarded to N.T. (Principal Investigator) and A.J.B. (Co-Investigator). J.E.C. was part-funded by a grant from the Medical Research Council (G0800973) awarded to N.T. (Principal Investigator) and A.J.B. (Co-Investigator).

## Author contributions

C.J.M., J.E.C., D.N., B.J. and L.E.B. acquired and interpreted data. A.J.B. and N.T. interpreted data and obtained funding. C.J.M., A.J.B. and N.T. contributed to study concept and design, and drafted and critically revised the manuscript.

## Additional information

**Competing interests:** The authors declare no competing financial interests.

