## [Peer Review File · Nature Communications]

Reviewers' Comments:

Reviewer #2 (Remarks to the Author)

This is an exciting, novel, interesting and important study. Interpretation of the data is however challenging because the transplanted cells not only restore the nitrergic relaxation in the colon of nNOS^{-/-} mice, but they also result in a large increase in the amplitude of basal contractions. This raises the possibility that the increased amplitude of the basal contractions is due to an inflammatory response, and that the rescue of GI transit and fecal output are due to increased basal contraction amplitude and hence possibly inflammation. The transplanted cells restore the ICC networks in nNOS^{-/-} colon, but the mean basal contraction amplitude in transplanted nNOS^{-/-} animals is 3 fold higher than C56Bl/6J.

Although interpretation is difficult, I am very enthusiastic about this study. I am confident it will generate a lot of interest and be influential in the field.

Specific points

1. The authors show that nNOS⁺ neurons are present within neurospheres in vitro. Do the neurospheres contain other types of neurons such as serotonin, CGRP, ChAT?
2. It is not clear whether the YFP⁺ structures in the proximal colon are cells or nerve fibers. Are the structures in the proximal colon TuJ1⁺? If not, do the YFP⁺ structures in the proximal colon express glial markers? Is there continuity between the network of YFP⁺ cells at the transplant site in the distal colon and most distant YFP⁺ structures in the proximal colon?
3. Are the mean values for EFS-induced relaxation for transplanted nNOS^{-/-} mice statistically different from C57BL/6J?
4. In regard to the increase in basal contraction amplitude in the transplanted nNOS^{-/-} group: Have you examined whether anti-inflammatory drugs reduce the amplitude of the basal contractions in transplanted nNOS^{-/-} animals? Is there an increase in basal contractions in the colon of C57BL/6J mice that contain transplanted cells?

Minor:

1. As the Materials and Methods are online, state that the donor mice were P2-P7 in the first paragraph of the Results.
2. Line 120: nNOS⁺ neurons are reported to be absent from the distal colon. I presume nNOS⁺ neurons are completely missing in these mice, not just in the distal colon.
3. In Fig. 2e, place the "200 bp" label for the ladder on the side of the gel, not over the top of the gel.
4. Enlarge figures 2f and g and show each channel as separate images, as it is impossible to determine if the BRDU⁺ nuclei are present in TuJ1⁺ cells in the images shown. Fig. 1d-i are also very small.

Reviewer #3 (Remarks to the Author)

This study by McCann et al examines the extent to which transplanted enteric neuronal stem cells (ENSC) can restore colonic motor function in nNOS deficient mice. Nitric oxide is a major inhibitory neurotransmitter in the gut and these knockout animals have a deficit in inhibitory neuromuscular transmission, which is manifested as delayed colonic transit. In this study these animals are used as a model of enteric neuropathy such as seen in patients with slow transit constipation. The rationale for the study was that by replenishing nitroergic neurons via ENSC transplantation that

colon motor function would be restored. This indeed proved to be the case, which is a striking observation and one that potentially has clinical relevance in treating human disease. However, an intriguing observation is the extent to which ENSC transplantation influences the numbers of ICCs within the gut wall. Networks of these non-neuronal cells are located in different locations in the gut wall where they play a critical role in pacemaker activity and in neuromuscular neurotransmission. Thus the rescue of colonic motor function appears to depend upon both cell-autonomous and non cell-autonomous mechanism.

The authors have used immunocytochemistry and cell imaging techniques to track ENSC as they colonise, proliferate and integrate within the mouse gut using a transgenic approach to tag ENSC with YFP. Some of these ENSC express nNOS forming a network of neurons that extends from the transplant site in the distal colon into more proximal regions. The implication is that these are inhibitory motoneurons that restore neuromuscular transmission but was there any further attempt to phenotype these neurons into interneurons/motoneurons?

The consequences of restored inhibition were addressed in a number of functional readouts. These consistently demonstrate rescued responses in both NANC mediated muscle tone, spontaneous contractile activity and colonic (but not gastric or small bowel) transit. In Fig 3 a rebound contraction is a striking feature of the off-response to electrical stimulation. Was there any attempt to quantify this?

Fig 4 demonstrates an increase in contraction amplitude but not frequency in ENSC transplanted colon. TTX is used to explore the neurogenic/myogenic contribution to these motor patterns, which because they are not affected by TTX suggests myogenic mechanisms. This data is not shown but I suggest might be relevant to the discussion since it appears from this example that TTX leads to an increase in contractile amplitude (assuming these are before and after traces). If this is the case it suggests that there is ongoing inhibitory neuronal tone in transplanted tissue that is released upon addition of TTX. In supplementary fig 3, this does not appear to be the case for proximal colon which might indicate subtle regional differences in the response to ENSC transplantation.

Data in Fig 6 show that ICC number in proximal and distal colon is increased after ENSC transplantation. Is it possible in these confocal images to distinguish ICC in different layers. It would be particularly interesting to see if there were changes in the deep muscular plexus where ICCs play a role in neuromuscular transmission. It would also be interesting to know if other cells in this region were affected, particularly the PDGFR α ⁺ cells, since this might have a bearing on other aspects of inhibitory transmission.

The fundamental question that this study poses is the extent to which improved function is as a consequence of restored neural innervation or ICCs. ICC deficient mice (w/wv mutants) have a deficit in inhibitory neurotransmission despite an intact innervation. Could the transplanted neurons provide a neurotrophic factor that maintains ICC differentiation? The obvious candidate would be stem cell factor and I wonder if the authors have been tempted to look at this in their PCR analysis?

Other comments

YFP and GFP seem to be used interchangeably in the figures and legends. Is this correct?

From the list of abbreviations I was expecting to see attempts to further characterize the nNOS⁺ neurons based upon the presence or absence of other co-transmitters.

Fig 2. The reference to panels e,f and g are incorrect.

signed David Grundy

Below we have addressed each of the reviewer's comments and where applicable, changes have been made to the manuscript we have included to appropriate line number and figure reference.

Reviewer #2 (Remarks to the Author):

Specific points

1. The authors show that nNOS+ neurons are present within neurospheres *in vitro*. Do the neurospheres contain other types of neurons such as serotonin, CGRP, ChAT?

Thank you for this comment. Neurospheres have been extensively characterised in our previous studies. In earlier published work from our group (Binder et al, 2015) neurospheres *in vitro* were shown to contain the neuronal subtype markers CGRP, NPY, nNOS, VIP, and the glial cell markers GFAP and S100. A further study from our group (Cooper et al, 2016) characterised neuronal phenotypes of transplanted ENSC within the wildtype mouse colon *in vivo*. Here we demonstrated that neurosphere derived cells have the ability to form the neurons (TuJ1) including inhibitory neurons (nNOS and VIP) and excitatory neurons (ChAT and Calbindin). The selection of ENSC used in the present study was identical to that of our previous work, hence we assume that the neurospheres used have similar characteristics including other types of neurons.

2. It is not clear whether the YFP+ structures in the proximal colon are cells or nerve fibers. Are the structures in the proximal colon TuJ1+? If not, do the YFP+ structures in the proximal colon express glial markers? Is there continuity between the network of YFP+ cells at the transplant site in the distal colon and most distant YFP+ structures in the proximal colon?

Thank you for these comments. To further add to this characterisation we have included a number of additional figures within the manuscript.

Using whole colon confocal microscopy we have demonstrated integration of GFP+/TuJ1+ cells within myenteric ganglia along the length of the colon,

including proximal segments. Representative images of this are presented in Supplementary Figure 3 and Supplementary Movie 2 and have been added to the results section (Line 134).

We have also addressed the query regarding continuity of transplanted networks in Supplementary Figure 2 with the inclusion of a transplanted cell map. Using standard microscopy we were able to show that networks of transplanted cells formed continuous networks extending almost 11mm along the length of the colon, which has also been added to the results (Line 126). Such quantification is technically challenging given artefacts within the tissue after dissection etc hence we likely underestimate the full continuity of a “network.”

Are the mean values for EFS-induced relaxation for transplanted nNOS^{-/-} mice statistically different from C57BL/6J?

Thank you for this comment. We can confirm that the EFS-induced relaxation for transplanted nNOS^{-/-} mice is statistically different from C57BL/6J. This has been added to the figure and the results section (Line 175).

3. In regard to the increase in basal contraction amplitude in the transplanted nNOS^{-/-} group: Have you examined whether anti-inflammatory drugs reduce the amplitude of the basal contractions in transplanted nNOS^{-/-} animals? Is there an increase in basal contractions in the colon of C57BL/6J mice that contain transplanted cells?

Thank you for this comment. We had previously stated in the manuscript that after careful analysis, evidence of inflammation was not observed within any of the colonic tissues. This data was presented as H&E sections. To further add to this we have now included representative images of colonic specimens from C57BL/6J, nNOS^{-/-}, sham-operated nNOS^{-/-}, and transplanted nNOS^{-/-} mice demonstrating similar gross morphology and the absence of an inflammatory response in Supplementary Fig7a. As we could not observe any evidence of an inflammatory response we do not feel that the inclusion of anti-inflammatory drugs in organ bath experiments would provide additional insights into the mechanisms of increased contractile amplitude. To date, we not assessed contractile activity in transplanted C57BL/6J colon with previous work looking at functional integration of transplanted mouse (Cooper et al, 2016) and human ENSC (Cooper et al, 2017) in C57BL/6J tissues via calcium imaging. While this is an interesting question we feel that this is outside of the scope of the present study, which looks specifically at transplantation, and functional rescue of the nNOS^{-/-} colon.

Minor:

1. As the Materials and Methods are online, state that the donor mice were P2-P7 in the first paragraph of the Results

Thank you for this. This has now been added to Results section (Line 103)

- as suggested.
2. Line 120: nNOS⁺ neurons are reported to be absent from the distal colon. I presume nNOS⁺ neurons are completely missing in these mice, not just in the distal colon.
Thanks for this. This is correct and we can confirm that nNOS⁺ neurons are completely absent in these mice. We have corrected line 115 to read “nNOS^{-/-} mice display complete loss of nNOS⁺ neurons in the colon (**Fig. 1b**)” to avoid confusion and compliment the previous statement re Wildtype colon.
 3. In Fig. 2e, place the “200 bp” label for the ladder on the side of the gel, not over the top of the gel.
Thanks for this. This has been corrected in Figure 2e and removed to side
 4. Enlarge figures 2f and g and show each channel as separate images, as it is impossible to determine if the BRDU⁺ nuclei are present in TuJ1⁺ cells in the images shown. Fig. 1d-i are also very small.
Thank you for this. We have now corrected Figure 2 to separate the individual channels. Previous figure 2f now equals figs 2f-l, previous figure 2g now equals figs 2j-m.
We have also amended Figure 1d-l to increase increased size of the individual channels as suggested.

Reviewer #3 (Remarks to the Author):

The authors have used immunocytochemistry and cell imaging techniques to track ENSC as they colonise, proliferate and integrate within the mouse gut using a transgenic approach to tag ENSC with YFP. Some of these ENSC express nNOS forming a network of neurons that extends from the transplant site in the distal colon into more proximal regions. The implication is that these are inhibitory motoneurons that restore neuromuscular transmission but was there any further attempt to phenotype these neurons into interneurons/motoneurons?

Thank you for this comment. We have addressed this with the inclusion of an additional figure (Supplementary Figure 4) to demonstrate the incorporation of distinct neuronal phenotypes including transplant GFP⁺/Tuj1⁺ neurons showing characteristic interneuron and motor neuron morphology. An additional comment has been added to the results section (Line 138) to reflect this.

The consequences of restored inhibition were address in a number of functional readouts. These consistently demonstrate rescued responses in both NANC mediated muscle tone, spontaneous contractile activity and colonic (but not gastric or small bowel) transit. In Fig 3 a rebound contraction is a striking feature

of the off-response to electrical stimulation. Was there any attempt to quantify this?

Thank you for this comment. Quantification of this rebound contraction has now been added to the manuscript (Fig. 3f and in the results section from Line 177). The interpretation of these results is difficult and we have sought to address this in the discussion (Line 339). We have included references (Ward et al, 1992 and Franck et al, 1999) describing the proposed rebound contraction mechanisms however direct comparison between these studies are difficult given species and methodological differences (i.e. organ bath contractility vs intracellular microelectrode recordings) hence we have included a statement in the discussion that “further investigation is required to assess the functional interaction of transplanted ENSC at the cellular level” (Line 348).

Fig 4 demonstrates an increase in contraction amplitude but not frequency in ENSC transplanted colon. TTX is used to explore the neurogenic/myogenic contribution to these motor patterns, which because they are not affected by TTX suggests myogenic mechanisms. This data is not shown but I suggest might be relevant to the discussion since it appears from this example that TTX leads to an increase in contractile amplitude (assuming these are before and after traces). If this is the case it suggests that there is ongoing inhibitory neuronal tone in transplanted tissue that is released upon addition of TTX. In supplementary fig 3, this does not appear to be the case for proximal colon which might indicate subtle regional differences in the response to ENSC transplantation.

Thank you for this helpful observation, this had originally been overlooked. The traces presented are before and after TTX addition as you assume. Quantified analysis of the contractile amplitude before and after TTX has now been included in Supplementary Figure 6 and reference to the figure added to the results section (Line 215&234). Unfortunately, in performing this analysis it is clear that there is no difference between basal contractile amplitude before and after addition of TTX within the summary data. Due to this we feel that our original interpretation is valid, that transplantation of ENSC can lead to potential changes in underlying myogenic motility patterns.

Data in Fig 6 show that ICC number in proximal and distal colon is increased after ENSC transplantation. Is it possible in these confocal images to distinguish ICC in different layers. It would be particularly interesting to see if there were changes in the deep muscular plexus where ICCs play a role in neuromuscular transmission. It would also be interesting to know if other cells in this region were affected, particularly the PDGFR α + cells, since this might have a bearing on other aspects of inhibitory transmission.

Thank you for this comment. In reference to the methods “for cell-counting

experiments 5 Z-series scans (21x1µm optical sections including longitudinal muscle, myenteric plexus and circular muscle layers) were obtained in each region per animal. Z-series files were blinded prior to counting. Individual cells were identified with DAPI co-labeling and cell counts performed across each individual z-scan using the cell counter plugin (FIJI). Each z-series count (21 sections) was summed and average cell number was calculated as a mean of 5 Z-series scans per region in each animal.”

Imaging in this way did indeed allow for isolation of individual ICC populations however given the low numbers of cells imaged at high power (40X) within the individual (longitudinal and circular) muscle layers these did not reach significance and hence these numbers were incorporated into the total ICC count.

Unfortunately, to the best of our knowledge ICC at the level of the deep muscular plexus are restricted to small intestinal tissue and so could not be analysed in the current study. It would however be interesting to study the effect of transplantation in the small intestine and the effects on ICC-DMP.

The point raised regarding the PDGFRalpha+ cells is indeed interesting. Preliminary experiments suggested that as opposed to ICC, which were reduced in nNOS-/- colon, PDGFRalpha+ cells did not appear to be affected therefore we felt there was little rationale to investigate this within the scope of this current study. Saying that, we are currently investigating the role of PDGFRalpha+ cells as part of a larger characterisation study to understand the role of compensatory mechanisms when specific branches of neurotransmission are absent.

The fundamental question that this study poses is the extent to which improved function is as a consequence of restored neural innervation or ICCs. ICC deficient mice (w/wv mutants) have a deficit in inhibitory neurotransmission despite an intact innervation. Could the transplanted neurons provide a neurotrophic factor that maintains ICC differentiation? The obvious candidate would be stem cell factor and I wonder if the authors have been tempted to look at this in their PCR analysis?

Many thanks for this suggestion. We have examined the expression of SCF ligand in ENSC derived neurospheres and added this result to Figure 6g. Reference to this data has also been included this in the results (Line 317) and discussion (Line 465). In addition, we have added several references (Torihashi et al,1996; Young et al, 1998) to describe expression of SCF in enteric neurons as part of this expanded discussion.

Other comments

YFP and GFP seem to be used interchangeably in the figures and legends. Is this correct?

Thanks for this comment. YFP is used when referring to endogenous YFP

fluorescently labeled donor cells (non-immunolabeled). GFP is used for immunohistological experiments in which donor cells have been labeled with anti-GFP primary antibody and an Alexa Fluor 488 secondary antibody. We have amended the manuscript for continuity.

From the list of abbreviations I was expecting to see attempts to further characterize the nNOS+ neurons based upon the presence or absence of other co-transmitters.

Thank you for pointing this out. Apologies, this was an editing error on our part. The original abbreviation list was not amended from a draft manuscript, which described our previous work characterising neurospheres in greater detail (see above comment for reviewer 2). In shortening the manuscript for submission to Nature Communications these details were removed however the abbreviation list was not amended to reflect this. This has been amended in the revised submission.

Fig 2. The reference to panels e,f and g are incorrect.

Thank you for this comment. This has been amended in the main body of the manuscript and figure legend to reflect the revised figure, which similarly addresses the comments of reviewer 2.

Reviewers' Comments:

Reviewer #2:

Remarks to the Author:

The authors have satisfactorily addressed my questions and concerns. This is a novel and important study.

Reviewer #3:

Remarks to the Author:

The authors have undertaken a thorough revision of their manuscript including the addition of new data. The new data answers the questions I raised in my initial review. The authors have also dealt with the minor issues I raised.

Response to reviewer's comments

Reviewer #2 (Remarks to the Author):

The authors have satisfactorily addressed my questions and concerns. This is a novel and important study.

Reviewer #3 (Remarks to the Author):

The authors have undertaken a thorough revision of their manuscript including the addition of new data. The is new data answers the questions I raised in my initial review. The authors have also dealt with the minor issues I raised.

Many thanks for these comments. We would like to state our appreciation of the constructive comments during reveiw of our manuscript. We feel that addressing these comments has added strength to the overall study and manuscript.